# Association of Antihyperglycemic Therapy with Risk of Atrial Fibrillation and Stroke in Diabetic Patients

**DOI:** 10.3390/medicina55090592

**Published:** 2019-09-15

**Authors:** Cristina-Mihaela Lăcătușu, Elena-Daniela Grigorescu, Cristian Stătescu, Radu Andy Sascău, Alina Onofriescu, Bogdan-Mircea Mihai

**Affiliations:** 1Diabetes, Nutrition and Metabolic Diseases, “Grigore T. Popa” University of Medicine and Pharmacy, 700115 Iași, Romania; 2“Sf. Spiridon” Emergency Hospital, 700111 Iași, Romania; 3Internal Medicine, “Grigore T. Popa” University of Medicine and Pharmacy, 700115 Iași, Romania; 4“George I.M. Georgescu” Cardiovascular Diseases Institute, Cardiology Department, 700503 Iași, Romania

**Keywords:** diabetes mellitus, atrial fibrillation, stroke, metformin, thiazolidinediones, GLP-1 receptor agonists, SGLT-2 inhibitors

## Abstract

Type 2 diabetes mellitus (DM) is associated with an increased risk of cardiovascular disease (CVD). Atrial fibrillation (AF) and stroke are both forms of CVD that have major consequences in terms of disabilities and death among patients with diabetes; however, they are less present in the preoccupations of scientific researchers as a primary endpoint of clinical trials. Several publications have found DM to be associated with a higher risk for both AF and stroke; some of the main drugs used for glycemic control have been found to carry either increased, or decreased risks for AF or for stroke in DM patients. Given the risk for thromboembolic cerebrovascular events seen in AF patients, the question arises as to whether stroke and AF occurring with modified incidences in diabetic individuals under therapy with various classes of antihyperglycemic medications are interrelated and should be considered as a whole. At present, the medical literature lacks studies specifically designed to investigate a cause–effect relationship between the incidences of AF and stroke driven by different antidiabetic agents. In default of such proof, we reviewed the existing evidence correlating the major classes of glucose-controlling drugs with their associated risks for AF and stroke; however, supplementary proof is needed to explore a hypothetically causal relationship between these two, both of which display peculiar features in the setting of specific drug therapies for glycemic control.

## 1. Introduction

Cardiovascular disease (CVD) is the main cause of morbidity and mortality in type 2 diabetes patients. The increased cardiovascular risk seen in diabetic patients cannot be mitigated with a monofactorial intervention of plasma glucose control, requiring a multi-factorial control of all cardiovascular risk factors [1,2]. Some of the newer classes of antihyperglycemic drugs have the potential to improve other risk factors beyond glycemic levels, and to protect against major cardiovascular events. Hence, the presence of CVD has become one of the key decision factors in the international guidelines counseling the choice of second-line antidiabetic medication after metformin [3].

Among all potential clinical forms of diabetes-associated obstructive artery disease, cerebrovascular disease is a serious condition, inducing major disabilities and a shortened life span. In a large meta-analysis of 102 prospective studies, diabetes mellitus was associated with a 2.27-fold increase in the risk for ischemic stroke when compared with a non-diabetic status [4].

Accumulating clinical evidence also seems to connect diabetes mellitus with an increased risk for atrial fibrillation (AF) [5]. Diabetes mellitus may induce structural and electrical alterations of the left atrium (deposition of advanced end-glycation products and connexin-mediated fibrosis), and stimulate the production of pro-coagulant factors (von Willibrand factor, soluble P-selectin, and other molecules exerting pro-inflammatory and pro-oxidative actions or favoring platelet activation and aggregation) [6]. All these changes promote clotting in the left atrial appendage and subsequent thromboembolism [6].

In a turning point in diabetes-related clinical research, several older or newer drugs used to control glycemic values in diabetic patients were recently shown—mostly in observational studies, post-hoc analyses of the major trials, or various meta-analyses—to display different levels of risk for either AF or stroke [7,8]. Such evidence exists for all classes of antidiabetic drugs included in the major international guidelines [3,7,8]. These drugs are summarized in Table 1. The body of evidence accumulating for each of these two new facets of antidiabetic medications is continuously increasing, and may represent far more than a random coincidence, even though no studies have been drafted to investigate a specific cause-effect relationship between the AF and stroke risks associated with use of various antihyperglycemic agents. Therefore, the aim of the present review is to gather, for the first time in the literature, the current knowledge on the risks of each of the antihyperglycemic drugs advised by current guidelines for both AF and stroke, raising the question as to whether they are causally interconnected.

## 2. Antihyperglycemic Drugs, Atrial Fibrillation and Stroke

Recently published research has frequently depicted various classes of antihyperglycemic agents as being associated with modified levels of risk for either AF or stroke. Stroke episodes in AF patients frequently have a thromboembolic nature; hence, the question arises as to whether a specific risk for AF in one or the other of the antidiabetic drugs would reflect an accordingly modified risk for cerebral thromboembolism, and thus stroke. Unfortunately, the major clinical trials have not yet distinguished between the ischemic or hemorrhagic nature of stroke episodes, and least of all, between the atherothrombotic or thromboembolic etiology of ischemic strokes [9,10]. In the absence of dedicated studies using electrocardiogram (ECG) technologies to monitor the heart rhythm, a high number of asymptomatic AF and/or paroxysmal, recurrent episodes of AF may go unrecognized; this may underlie the inconstant associations between diabetes and incidences of AF or stroke seen in clinical studies, especially those not reporting AF as a specific outcome [11]. We searched Medline and Scopus databases using the logical string “atrial fibrillation” OR “stroke” AND “antihyperglycemic” AND “diabetes” to identify these key terms in the title or abstract of English-written articles published before June 2019. Clinical studies or trials, meta-analyses, and systematic reviews focusing on human subjects were selected. After eliminating duplicates, this initial search returned 14 results. We screened all titles and abstracts to select papers that could be considered relevant to the aim of our review. This operation led to a further reduction to only 11 titles. A second search using the same algorithm and replacing the key term of “antihyperglycemic“ with “insulin“ OR “metformin” OR “sulfonylurea (SU)” OR ”thiazolidindione (TZD)” OR ”dipeptidyl peptidase-4 (DPP-4) inhibitor” OR ”glucagon-like peptide-1 (GLP-1) receptor agonist” OR ”sodium-glucose cotransporter-2 (SGLT-2) inhibitor” issued 28 supplementary papers, which were also included in our review. When potential mechanistical explanations were useful, we also referred to other relevant review papers, selected by the same two search algorithms; as an only exception, we included a case report which filled a gap in an area of scarce evidence. The following sections summarize the data related to the risk of AF and stroke for each class of antihyperglycemic agents.

### 2.1. Insulin

In a case-control study on Taiwan registries, insulin therapy was associated with a higher risk of new-onset AF in diabetic patients than with other antihyperglycemic medications [12]. Among patients in the PREvention oF thromboembolic events—European Registry in Atrial Fibrillation (PREFER in AF) registry, insulin users, but not diabetic patients treated with non-insulin antihyperglycemic drugs, were shown to have a higher risk of stroke compared with non-diabetic individuals [13]. In a Medicare analysis on 798,592 AF patients, insulin-requiring diabetic subjects also had a higher risk of stroke than diabetic patients not requiring insulin therapy or non-diabetic individuals; use of insulin therapy was associated in this registry study with an attenuation in the efficacy of anticoagulant drugs [6]. However, the association between insulin therapy and this pro-arrhythmic status may be biased by the longer duration of type 2 diabetes usually seen in patients treated with insulin. Such subjects may have experienced years of suboptimal glycemic control on other non-insulin therapies, and may have had the time to develop significant comorbidities [12]. The real possibility exists that hyperinsulinism (either due to insulin resistance or, in this case, having an iatrogenic component) may be associated with an increased anti-fibrinolytic status, as insulin stimulates the Plasminogen Activator Inhibitor-1 (PAI-1) production in adipocytes [14].

### 2.2. Metformin

In a cohort study on 645,710 Taiwan patients, monotherapy with metformin was associated with a 19% reduction in the risk of AF compared with the use of other antihyperglycemic medications during a 13-year follow-up. Metformin users had the lowest AF incidence rates in the first two years after diagnosis, but the protective effect tended to fade afterward [15]. Possible explanations accounting for the favorable effect of metformin include its actions on adenosine monophosphate-activated kinase, and the drug-induced reduction of the oxidative stress and the myolysis in the atrial tissue [15,16]. The loss of its protective effect over time may be underlain by the progressive deterioration of β-cell function typically observed in type 2 diabetes, which may lead to a worsened glycemic control, or by the gradual remodeling of the atrial wall [15]. In the above-mentioned case-control study, also originating from Taiwan registries, biguanides, of which metformin is the main representative today, were also associated with a lower risk of developing AF [12].

Current evidence suggests that metformin also has a protective effect against ischemic stroke, even though specific outcome studies analyzing a potential cause–effect relationship between the protective role of metformin against AF development and the rate of thromboembolic events are lacking in the medical literature. The results of the United Kingdom Prospective Diabetes Study (UKPDS) suggested that intensive blood glucose control with metformin, compared with the use of sulfonylureas or insulin, significantly reduced the risk of stroke [17]. After a four-year follow-up, the administration of metformin within the antihyperglycemic therapy was associated with a 54% reduction in the risk of stroke, with the best results observed in the highest risk patients [18].

### 2.3. Sulfonylureas

Most researchers analyzing the risk of AF development have considered SU therapy as only a control to report comparative AF outcomes of other antidiabetic medications. Among the few studies making an exception, the previously mentioned Taiwan case-control study found SUs to not be associated with an increased risk of new-onset AF [12].

Research on the stroke risk associated with SU use generally precedes the publication of most studies using these drugs as an active comparator for other medications’ AF risk. This class of hypoglycemiant drugs acts on the SU receptor (SUR) unit of the ATP-sensitive potassium channels. In normal conditions, these ionic channels may play a protective role against neuronal ischemia. SUs were therefore feared by some authors to inhibit this neuroprotective mechanism, and thus to increase the risk of stroke [19,20,21]. Initial results of clinical studies were contradictory, varying between reports of potential benefits [22], neutral effects [23], or even detrimental effects [24]. A subsequent meta-analysis of 27,705 diabetic patients from 17 trials found SUs to be associated with a higher relative risk for stroke than other antihyperglycemic drugs administered for glycemic control [20].

### 2.4. Thiazolidinediones

Thiazolidinediones are insulin sensitizers acting primarily on the peroxisome proliferator-activated receptor (PPAR)-γ and, in the case of pioglitazone, also exerting a weak agonist activity on PPAR-α. Their action on these nuclear receptors is associated with anti-inflammatory and anti-oxidant benefits, potentially due to favorable effects on Transforming Growth Factor (TGF)-β, Tumor Necrosis Factor (TNF)-α, Atrial Natriuretic Peptide (ANP), superoxide dismutase (SOD), malonyldialdehyde, nicotinamide adenine dinucleotide phosphate (NADPH) oxidase subunits, or voltage-dependent calcium channels [25]. Reports of an increased risk of hydro-saline retention, heart failure, and cardiovascular events seen with rosiglitazone [26,27] drastically limited their use in diabetic patients. As a direct effect of these reports, regulatory agencies subsequently requested proof of cardiovascular safety for the newer generations of antihyperglycemic drugs by means of dedicated trials.

These conflicting features of TZDs led to research on their association with atrial fibrillation and stroke. In an observational study on 12,605 patients with insulin-naïve type 2 diabetes, the risk of developing AF was reduced by 31% after a five-year follow-up in patients treated with TZD [28]. In another smaller observational study following the arrhythmic outcomes of catheter ablation, pioglitazone was also reported to be associated with a reduced risk of post-procedural AF [29]. A better recovery to sinus rhythm was reported in isolated cases of patients with paroxysmal AF and diabetes who received rosiglitazone [30]. The use of TZDs was associated with a lower risk of developing AF in the Taiwan case-control study that was previously mentioned [12]. A large cohort study of 108,624 diabetic, AF-free Danish patients, treated with either metformin or sulfonylureas as first-line antihyperglycemic therapy, showed a 24% risk reduction in the incidence of AF when TZDs were used as a second-line drug for glycemic control, compared with other antidiabetic drugs [31]. Post hoc analyses on the incidence of AF in the PROactive (PROspective pioglitAzone Clinical Trial In macroVascular Events) and BARI 2D (Bypass Angioplasty Revascularization Investigation 2 Diabetes) trials did not show significant differences in the number of patients developing AF [32,33]. However, neither of these two randomized studies were designed to include AF between their specific endpoints, so they did not systematically search for its existence using any ECG-monitoring device. The number of patients receiving TZDs who developed AF was lower than their counterparts in both studies [32,33]. A meta-analysis including 130,854 patients from three randomized clinical trials and four observational studies found a 30% reduction in the AF risk in patients treated with TZD, with significantly reduced incidences of both new-onset AF and recurrent AF [34]. In this meta-analysis, results were observed predominantly with pioglitazone, but not with rosiglitazone, and were driven by the data in the observational studies, as the pooled analysis of the results from the three randomized clinical trials showed no statistical differences in the AF incidence [34].

Similar to the case of metformin, no specific evidence links the potentially protective role of TZDs against AF and the effect of these drugs on the risk of cerebrovascular events. However, some data indicate a real possibility that TZDs have the ability to protect diabetic patients against stroke development. In another sub-analysis of the PROactive study, the risk for fatal or non-fatal stroke was significantly reduced with pioglitazone in type 2 diabetes patients with a history of previous stroke, but not in those without a history of cerebrovascular events [35]. In the Insulin Resistance Intervention after Stroke (IRIS) trial, performed in non-diabetic but insulin-resistant patients with a history of stroke or transient ischemic attack, pioglitazone was able to lower the risk for recurrent stroke or myocardial infarction compared with placebo therapy [36]. Finally, a meta-analysis on three randomized controlled trials, incorporating 4980 subjects with previous stroke and either insulin resistance, prediabetes, or type 2 diabetes mellitus, found the use of pioglitazone to be associated with a 32% lower risk of stroke recurrence compared with a placebo [37].

### 2.5. DPP-4 Inhibitors

In a cohort study on 90,880 patients with type 2 diabetes previously treated with metformin as a first-line antihyperglycemic drug, the add-on of DPP-4 inhibitors (mostly sitagliptin) as a second-line therapy was found to be associated with a lower risk of AF development than the use of other drugs (mainly SUs) as the second antidiabetic medication [38]. The use of DPP-4 inhibitors was associated with neither an increased nor a decreased risk of new-onset AF in the case-control study on Taiwan registries that was previously mentioned [12].

These positive or neutral results on AF risk raised the question of potentially protective effects of DPP-4 inhibitors against stroke. In another longitudinal observational Taiwan study on 123,050 type 2 diabetes patients that were newly initiated on oral antidiabetic drugs, the use of DPP-4 inhibitors was associated with a lower risk for ischemic stroke compared with meglitinides or insulin; however, their risk for stroke was comparable to that observed in metformin users, and higher than the risk observed in patients treated with pioglitazone [39]. None of the cardiovascular outcome trials with DPP-4 inhibitors identified a reduced risk for stroke with any of these medications [40,41,42,43,44]. When a meta-analysis was performed on 19 small randomized trials and the first three cardiovascular outcome trials with DPP-4 inhibitors that were published, a non-significant trend toward protection against stroke was found, but this trend disappeared when only the cardiovascular outcome trials were introduced into another pooled analysis [45]. Likewise, a meta-analysis of the five cardiovascular outcome trials with DPP-4 inhibitors available at the end of 2018 showed a neutral effect on the risk for stroke, similar to the profile of safety, but showed a lack of benefits in terms of the risk for myocardial infarction, cardiovascular death, or heart failure [46]. Similar to the case with other drugs, none of the cardiovascular outcome trials or meta-analyses with DPP-4 inhibitors published so far have differentiated between stroke events of hemorrhagic or ischemic origin, least of all between atherothrombotic or thromboembolic events.

### 2.6. GLP-1 Receptor Agonists

A side effect of GLP-1 receptor agonists includes a moderate increase in heart rate [47], which may be due to either an effect of the direct stimulation of the GLP-1 receptor found on sino-atrial cells, or a compensatory response to the relative lowering of blood pressure levels seen with GLP-1 receptor agonists [48,49]. Acknowledgement of this effect on the heart rate led to concerns that GLP-1 receptor agonists may be associated with a higher risk for AF, especially after a pooled analysis of the phase 2b and phase 3 trials in the Albiglutide and cardiovascular outcomes in patients with type 2 diabetes and cardiovascular disease (Harmony Outcomes) program with albiglutide showed a statistically significant increase in the AF incidence with this drug [50]. However, the cardiovascular outcome trials with lixisenatide, liraglutide, or semaglutide found no differences in the AF incidence between any of the active drugs and the placebo comparator [51,52,53]. As cardiovascular outcome trials are specifically designed to follow major cardiovascular events, it is plausible to think that an AF episode—even though not counting as a pre-defined endpoint—should be more recognized in such studies than in trials with metabolic outcomes, to therefore offer a better statistical accuracy. Since these three cardiovascular outcome trials included patients with pre-existing cardiovascular disease, it is also presumable that such subjects would be treated with β-blockers, thus reducing the probability of AF occurrence and reducing the number of cases below the limit of statistical significance. Subsequently, a meta-analysis of all trials available in 2017 with GLP-1 receptor agonists (including studies with albiglutide, but also with exenatide, lixisenatide, liraglutide, dulaglutide, and semaglutide) showed no increase in the risk of AF with these drugs [54].

However, aside from speculations about the risks of AF, GLP-1 receptor agonists are definitely not associated with a higher risk for stroke. All GLP-1 receptor agonists developed from the human GLP-1 backbone (liraglutide, injectable semaglutide, albiglutide, and dulaglutide) are able to lower the risk for the composite outcome of major cardiovascular events (cardiovascular death, non-fatal myocardial infarction, and non-fatal stroke) [53,55,56,57]. When endpoints included in the composite outcome were analyzed separately in each of these trials, liraglutide and albiglutide demonstrated non-significant differences opposite to the placebo in terms of the risk of stroke [55,56], whereas injectable semaglutide showed a significant 39% reduction [53], and dulaglutide was associated with a 24% reduction in the calculated risks for non-fatal stroke [57]. In a previously mentioned meta-analysis, including, in this case, the four cardiovascular outcome trials with GLP-1 receptor agonists available at the end of 2018, this class of drugs was associated with a 13% reduction in the risk for non-fatal stroke, even if atherothrombotic, thromboembolic, and/or hemorrhagic events were not differentiated [46].

### 2.7. SGLT-2 Inhibitors

SGLT-2 inhibitors exert their actions by inhibiting the active reabsorption performed by this specific co-transporter of sodium and glucose at the level of the proximal convoluted tubule. As a result, glucose, sodium, and water are lost in the final urine, lowering blood pressure and blood glucose levels, and creating a negative energy balance that induces weight loss. Based on these direct effects on multiple cardiovascular risk factors, but also on other adjunctive metabolic actions, SGLT-2 inhibitors seem able to lower the cardiovascular risk in diabetic patients. In the dedicated cardiovascular outcome trials, empagliflozin and canagliflozin were shown to reduce the progression to the composite outcome of major cardiovascular events [58,59], whereas dapagliflozin reduced the risk for the composite outcome of cardiovascular death and hospitalization for heart failure [60]. Currently, no research on the risk of AF development with any of the SGLT-2 inhibitors has been published, but a sub-analysis of the Empagliflozin Cardiovascular Outcome Event Trial in Type 2 Diabetes Mellitus Patients-Removing Excess Glucose (EMPA-REG OUTCOME) acknowledged a slightly increased incidence of stroke in the empagliflozin treatment group, even though not reaching statistical significance [61]. A subsequent meta-analysis of 57 studies using seven different approved or unapproved SGLT-2 inhibitors reported a 30% higher risk of non-fatal stroke [62]. Hypothetical explanations attribute this negative effect either to chance or to the relative increase in hematocrit, leading to a higher blood viscosity, as these agents exert an effect of osmotic diuresis [63]. However, another meta-analysis of trials with SGLT-2 inhibitors, this time including studies lasting at least 24 weeks and reporting at least one cardiovascular outcome, did not confirm an increased risk of stroke, thus assuring a reasonable level of cerebrovascular safety with this class of drugs [64]. The above-mentioned pooled analysis, including all three available cardiovascular outcome trials with SGLT-2 inhibitors, revealed no supplementary risk of stroke with SGLT-2 inhibitors compared with placebo comparators [46].

## 3. Conclusions

Current evidence supports the existence of a relationship between diabetes mellitus and an increased risk for atrial fibrillation and stroke. In these high-risk patients, several reports linking antidiabetic medications to modified risks for atrial fibrillation, stroke, or both, have been published in the last years. The most relevant of these results are summarized in Table 2.

The cause–effect relationship between the modified risk for atrial fibrillation of these drugs and cerebrovascular disease due to thromboembolic events has not yet been analyzed in studies with dedicated outcomes. However, depicting the ability of some specific antihyperglycemic therapies in reducing the risks for both atrial fibrillation and stroke as completely separate mechanisms would mean allowing the existence of slightly too much coincidental evidence. Trials searching for a potentially causal triangular relationship between antidiabetic drugs, risks for atrial fibrillation, and cerebral thromboembolism are needed to fill in a gap in evidence, and to potentially supplement the adaptation of the recommendations of current guidelines to prevent the negative outcomes of cardiovascular disease in diabetic patients as much as possible.

## Figures and Tables

**Table 1 medicina-55-00592-t001:** Classes of antihyperglycemic drugs included in current guidelines [3].

Drug	Mechanism of Action
Insulin	Activation of insulin receptor; various effects on multiple metabolic pathways
Metformin	Reduced insulin resistance, mostly by decreasing gluconeogenesis
Sulfonylureas (SU)	Insulin secretagogues by activation of SUR (SU receptor) unit of ATP-sensitive potassium channels
Thiazolidinediones (TZD)	Insulin sensitizers by the activation of peroxisome proliferator-activated receptor (PPAR)-γ
Dipeptidyl peptidase-4 (DPP-4) inhibitors	Inhibition of DPP-4 and subsequent conservation of native human GLP-1 in its active form
Glucagon-like peptide-1 (GLP-1) receptor agonists	Activation of GLP-1 receptor at high pharmacological concentrations
Sodium-glucose cotransporter-2 (SGLT-2) inhibitors	Inhibition of active reabsorption of glucose and sodium performed by SGLT-2 in the proximal convoluted tubule

**Table 2 medicina-55-00592-t002:** Summary of the main current evidence on the association of current antihyperglycemic drugs with risks of atrial fibrillation (AF) and stroke.

Drug	Risk for AF	Risk for Stroke
Insulin	Increased [12]	Increased [6,13]
Metformin	Reduced [12,15]	Reduced [17,18]
Sulfonylureas	Unchanged [12]	Reduced [22], unchanged [23], or increased [20,24]
Thiazolidinediones	Reduced [12,28,29,31,34] or unchanged [32,33]	Reduced [35,36,37]
DPP-4 inhibitors	Reduced [38] or unchanged [12]	Reduced [39] or unchanged [40,41,42,43,44,45,46]
GLP-1 receptor agonists	Increased with albiglutide [50], unchanged with semaglutide, liraglutide, and dulaglutide, or in meta-analyses [51,52,53,54]	Reduced in meta-analyses [46] and with semaglutide [53], unchanged with liraglutide, albiglutide, and dulaglutide [55,56,57]
SGLT-2 inhibitors	Data not available	Increased in some meta-analyses [62], unchanged in others [46,64]

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
