# Peer review of "Association of Antihyperglycemic Therapy with Risk of Atrial Fibrillation and Stroke in Diabetic Patients"

_medicina, 2019, doi:10.3390/medicina55090592_

Round 1

Reviewer 1 Report

The Authors provided a nice review on the association of antihyperglycemic therapy with risk of atrial fibrillation and stroke in diabetic patients.

The review is interesting and informative to a global audience. 

Reviewer 2 Report

In the current paper, LăcătuÈ™u and her collaborators reviewed the literature in search of any association between the anti-diabetic treatments/drugs and the risk of Atrial Fibrillation and stroke. The authors discussed the available data on the effect of seven large-scale recommended anti-diabetic drugs on the risk of developing Atrial fibrillation and stroke on the patients under treatment. Although the review brings together the up to date findings on this topic in one document, few major comments should be taken into consideration to make it scientifically sounds:

1- Most of the review is based on run-on sentences including many ideas which make the main sense lost in between. Please reformulate your manuscript using more simple sentences. I would recommend a thorough language revision by a native English speaker.

2- Adding figures or summarizing tables would make it easier to the readers to take the essential of the review.

3- It would be good if the authors add a sentence in the introduction explaining the choice of the discussed drugs.

Thank you.

Reviewer 3 Report

In the current manuscript, the authors presented the new evidence behind anti diabetic medications and the risk of thrombo-embolic stroke in diabetic patients. The paper is well written, updated references and well done.

Round 2

Reviewer 2 Report

The new version of the manuscript seems to be more consistent and comprehensible. Thanks to the authors for having answered to the main comments and having changed what's needed accordingly.